# ATTENTION TO MAMBA:
# A RECIPE FOR CROSS-ARCHITECTURE DISTILLATION

## ABSTRACT

State Space Models (SSMs) such as Mamba have become a popular alternative to Transformer models, due to their reduced memory consumption and higher throughput at generation compared to their Attention-based counterparts. On the other hand, the community has built up a considerable body of knowledge on how to train Transformers, and many pretrained Transformer models are readily available. To facilitate the adoption of SSMs while leveraging existing pretrained Transformers, we aim to identify an effective recipe to distill an Attention-based model into a Mamba-like architecture. In prior work on cross-architecture distillation, however, it has been shown that a naïve distillation procedure from Transformers to Mamba fails to preserve the original teacher performance, a limitation often overcome with hybrid solutions combining Attention and SSM blocks. The key argument from our work is that, by equipping Mamba with a principled initialization, we can recover an overall better recipe for cross-architectural distillation. To this end, we propose a principled two-stage approach: first, we distill knowledge from a traditional Transformer into a linearized version of Attention, using an adaptation of the *kernel trick*. Then, we distill the linearized version into an adapted Mamba model that does not use any Attention block. Overall, the distilled Mamba model is able to preserve the original Pythia-1B Transformer performance in downstream tasks, maintaining a perplexity of 14.11 close to the teacher's 13.86. To show the efficacy of our recipe, we conduct thorough ablations at 1B scale with 10B tokens varying sequence mixer architecture, scaling analysis on model sizes and total distillation tokens, and a sensitivity analysis on tokens allocation between stages.

## 1 INTRODUCTION AND MOTIVATION

Much of the development of natural language processing over the last decade can be directly attributed to the effectiveness of the Attention mechanism (Bahdanau et al., 2015; Vaswani et al., 2017) in generating rich, context-aware tokens representations, and unlocking parallel training. The power of Attention, however, comes at a computational cost scaling quadratically in the length of the input sequence $L$. The attempt to curb this requirement has triggered the development of a number of alternatives to Attention, which could retain linear complexity in $L$. Among these, some of the most successful are Linear Attention (Katharopoulos et al., 2020), RWKV (Peng et al., 2023), and State-Space Models (SSMs), particularly represented by Mamba (Gu & Dao, 2023; Dao & Gu, 2024).

On the one hand, the promise of faster inference times and reduced memory requirements provided by linear alternatives to Attention is undoubtedly appealing; on the other, their performance on downstream tasks still tends to fall short of that of Transformers, especially at scale. At the same time, research on Transformers is more mature, with a larger number of models available (Wolf et al., 2020), and considerable computational resources already spent into pretraining said models (Castaño et al., 2024). In light of this, instead of training SSMs from scratch, a promising direction is distillation (Hinton et al., 2015), which allows to directly leverage the knowledge embedded in readily-available pretrained Transformer models. Naïve direct distillation between Transformer and Mamba architectures, however, has shown to be challenging (Wang et al., 2024; Bick et al., 2024), and often failing to preserve teacher performance. In our work, we identify a critical missing piece: architectural alignment through principled initialization. Rather than forcing knowledge transfer across fundamentally different computational paradigms, we propose a two-stage bridging strategy (illustrated in Fig. 2) that exploits the mathematical connections between sequence mixers. We first

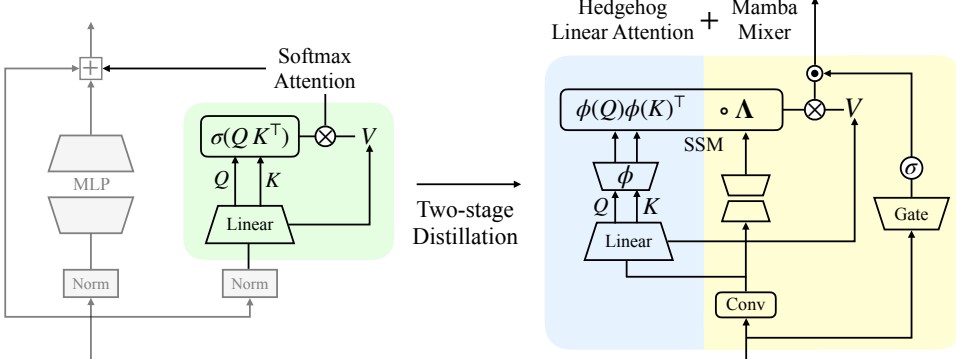

Figure 1: We propose a two-stage recipe to distill quadratic Softmax Attention (green) in a Transformer layer to a subquadratic Mamba-based Mixer module. Our sequence mixer (HedgeMamba) is a hybrid of a learned linear Attention (Hedgehog (Zhang et al., 2024) in blue) and Mamba Gu & Dao (2023) (yellow). Note that we keep the rest of the Transformer layer as it is from the teacher model (grey); we only swap Softmax Attention with our proposed HedgeMamba mixer.

distill standard Softmax Attention into Linear Attention, building up on the Hedgehog approach illustrated in Zhang et al. (2024). This is grounded in an application of the *kernel trick*, whereby the exponentials in the Attention scores computation are approximated by scalar products of specific features. The Linear Attention weights recovered after this first step are then used as an initialization for the Mamba parameters, and the whole model is further fine-tuned. The recipe is designed to guarantee effective knowledge transfer while limiting training cost to a fraction of the one used in pretraining the teacher Transformer. As teacher models, we consider the family of pretrained Pythia Transformers (Biderman et al., 2023), which we distill on an SSM-adaptation of their architecture. For the distillation procedure, we use data from the OpenWebText dataset (Gokaslan et al., 2019). The performance of our distillation recipe is measured both in terms of sheer perplexity, and on effectiveness on downstream tasks from `lm-eval-harness` (Gao et al., 2021). Our approach retains most of the teacher model's performance: for 1B models, the student reaches a perplexity of 14.11 (from the teacher's 13.86), with good overall scores on downstream tasks. We further establish the robustness of our approach through ablations over student architecture components, a scaling analysis on model size and total distillation tokens, and a sensitivity analysis on the token budget allocation between stages.

**Contributions**    Overall, the main contributions from our work are two-fold:

- We propose a novel method for cross-architecture distillation from a Transformer to a Mamba model. The method composes of two stages, whereby distillation is performed first from Attention to Linear Attention, and then onto Mamba, with the goal of favoring knowledge transfer between the two architectures.

- We evaluate the method effectiveness via extensive ablation, scaling and sensitivity studies. These are aimed both at refining the details of our distillation procedure, as well as verifying its robustness with respect to the available distillation budget.

## 1.1 PREVIOUS WORK

**Attention linearization**    Attention linearization techniques aim at simplifying the operations involved in the assembly and/or application of the Attention matrix, so that their computational complexity scales *linearly* (rather than *quadratically*) with the sequence length. Some methods proposed in the literature achieve this by directly modifying the structure of the Attention matrix, either by sparsifying it (Beltagy et al., 2020; Zaheer et al., 2021) or by reducing it to low-rank (Wang et al., 2020; Xiong et al., 2021). Most relevant for our work is a different approach, namely kernel-based Attention linearization (Katharopoulos et al., 2020; Choromanski et al., 2022; Peng et al., 2023; Qin et al., 2022; Peng et al., 2021). This line of research interprets the positive semi-definite

Attention matrix as a kernel application, which gets decomposed as dot products of feature vectors in high-dimensional space. Effectively, this allows to reduce Attention to a Recurrent Neural Network (RNN) application. The kernel-based approach is hence of particular relevance to us as it helps bridging the gap between our two targets architectures, namely Transformers and Mamba: the latter can in fact be interpreted as a specific RNN instantiation.

**State-Space Models**   State-Space Models are a specialization of RNNs relying on recurrence formulas which are purely linear in the hidden state. Imposing linearity has the advantage of rendering the computation of the recurrence relationship parallelizable along the input sequence during training, thus overcoming one of the main limitations of classical RNNs. The line of research analyzing the properties of SSMs has been particularly active (Gu et al., 2020; 2022a;b; Cirone et al., 2025), eventually producing the Mamba architecture (Gu & Dao, 2023), which has gained particular traction as a Transformer alternative. More recently, Dao & Gu (2024) has drawn a direct connection between Mamba and Linear Attention (equipped with a learnable causal mask). This, together with the performance showcased by Mamba, acts as main motivation for using the Mamba architecture as a representative for Linear Attention alternatives, and for adopting it in our student model definition.

**Cross-architecture distillation**   Knowledge Distillation (Hinton et al., 2015) is an established method for efficiently leveraging the knowledge embedded in an already-trained *teacher* model in order to accelerate the training of a *student* model, with a history of successful applications (Gou et al., 2021; Yang et al., 2024; Mansourian et al., 2025; Busbridge et al., 2025). While the focus of most of the available literature is on distilling a teacher into a (generally smaller) student of the same model class, in our work we are interested in distilling across two *different* architectures, with the purpose of reducing the computational complexity of Attention. The literature on quadratic-to-linear Attention distillation is much less developed in this sense, but the rise of Linear alternatives to Attention has recently sparkled interest in this specific area. For example, Scavenging Hyena (Ralambomihanta et al., 2024) distilled a Transformer model into a Hyena model (Poli et al., 2023) (but only for small scales <70M); SUPRA replaced softmax Attention directly with a linear application (Mercat et al., 2024); Wang et al. (2024) proposed distillation techniques for creating efficient hybrid Transformer-Mamba models; Mao (2022) simplifies distillation onto decaying fast weights, and (He & Garner, 2025) studies cross-architecture alignment strategies. More recently, MOHAWK (Bick et al., 2024; 2025) has attempted Transformer-to-Mamba distillation, proposing a three-stage recipe where the output of Attention and the SSM are progressively aligned before finetuning. A direct quantitative comparison with MOHAWK is confounded by fundamental differences in the underlying model architectures (our work is based on Pythia, while MOHAWK utilizes Phi as backbone), and the training sets (importantly, MOHAWK uses C4 which includes Book3, a dataset known to contain copyrighted material). However, a qualitative comparison of the methodologies is instructive. Both approaches aim to harness the expressivity of Mamba-like models, but differ significantly in their design and complexity. MOHAWK employs a complex three-stage training pipeline with distinct objectives and frozen modules for each stage. By contrast, our method proposes a two-stage recipe, theoretically grounded in the functional analogies between Transformers and SSMs, offering a more direct and computationally streamlined approach. Also relevant is LoLCATs (Zhang et al., 2025), which builds upon ideas from Hedgehog (Kasai et al., 2021; Zhang et al., 2024) (where softmax Attention is approximated via a learnable linear kernel) and aims at improving the architecture expressivity equipping it with windowed Attention and LoRA finetuning (Hu et al., 2021). While our work also builds on Hedgehog, LoLCats is unsuitable for direct comparison due scale differences and its instruction-finetuning loss, which cannot be applied to our pretraining-style setting.

## 2   PRELIMINARIES

In this section, we provide an overview of the target architectures for our distillation procedure, namely Transformer as teacher model and Mamba as SSM student model. We also highlight the connection between linearized forms of Attention and Mamba, which we leverage in developing our distillation recipe, as detailed in Sec. 3.

## 2.1 DESCRIPTION OF TARGET ARCHITECTURES

As representatives for the Transformer architecture, we consider models from the Pythia suite (Biderman et al., 2023). The suite contains publicly available models ranging in scale from 14M to 12B parameters, consistently trained following the same recipe. As target student model, we choose the Mamba architecture (Gu & Dao, 2023; Dao & Gu, 2024) which arguably represents the current state-of-the-art in SSM performance. For reference, schematics of the two architectures are provided in Fig. 4. It is worth pointing out that Mamba has been trained with the same tokenizer as Pythia, and for a similar number of steps.

At the highest level of abstraction, both the Transformer and Mamba architectures share a similar structure, which consists of interweaving two types of modules: one responsible for mixing tokens within a sequence (also called *sequence mixer*), the other for mixing components of each individual token embedding (generally performed by an MLP). Arguably, the most significant difference lies in the way the sequence mixing is performed in the two architectures, as described next.

**Attention** In the case of Transformers, it is the Self-Attention mechanism that is responsible for mixing the tokens embeddings sequence-wise. Its action on an input sequence $\boldsymbol{X} \in \mathbb{R}^{L \times d}$ (with $L$ being the sequence length, and $d$ the embedding dimension) is represented as

$$\boldsymbol{Y}_{\text{Attn}} \coloneqq \boldsymbol{A}_{\text{Attn}} \boldsymbol{V}, \qquad \text{with} \qquad \boldsymbol{A}_{\text{Attn}} \coloneqq \text{softmax}\left(\frac{\boldsymbol{Q}\boldsymbol{K}^{\mathsf{T}}}{\sqrt{d}}\right), \tag{1}$$

where $\boldsymbol{Q}, \boldsymbol{K}, \boldsymbol{V} \in \mathbb{R}^{L \times d}$ are linear transformations of $\boldsymbol{X}$, denoting queries, keys and values.

**SSM mixer** For Mamba, the sequence mixing is mainly performed by the SSM layer[1]. This reduces to unrolling a linear recurrence relationship in the form

$$\begin{aligned} \boldsymbol{h}_l &= \boldsymbol{\Lambda}_l \odot \boldsymbol{h}_{l-1} + \boldsymbol{B}_l \otimes \boldsymbol{X}_{l,:} & \text{for} \quad l = 1 \dots L, \\ \boldsymbol{Y}_{l,:} &= \boldsymbol{C}_l^{\mathsf{T}} \boldsymbol{h}_l, & \boldsymbol{h}_0 = \boldsymbol{0} \in \mathbb{R}^{N \times d}, \end{aligned} \tag{2}$$

for parameters $\boldsymbol{\Lambda}_l \in \mathbb{R}^{N \times d}$, and $\boldsymbol{B}_l, \boldsymbol{C}_l \in \mathbb{R}^N$, $N$ being the hidden state size. To highlight the similarity with (1), the solution to the above recurrence can be expressed in matrix form as

$$\boldsymbol{Y}_{\text{SSM}} \coloneqq \boldsymbol{A}_{\text{SSM}} \boldsymbol{X}, \qquad \text{with} \qquad [\boldsymbol{A}_{\text{SSM}}]_{i,j} \coloneqq \boldsymbol{C}_i^{\mathsf{T}} \prod_{k=i}^{j+1} \boldsymbol{\Lambda}_k \boldsymbol{B}_j, \tag{3}$$

which also indicates how, at training time, the SSM mixer can be applied in a parallel fashion along the sequence, similarly to Attention. The main particularity of Mamba, which sets it apart from other SSM models, lies in the fact that the recurrence parameters $\boldsymbol{\Lambda}_l, \boldsymbol{B}_l, \boldsymbol{C}_l$ all depend on the input $\boldsymbol{X}_{l,:}$. Indeed, this formulation makes it akin to Linear Attention alternatives, as we outline in the following.

**Linear Attention and SSMs** Linearized alternatives to Attention aim at turning the application of the Attention layer (1) from a quadratic- to a linear-complexity operation in $L$. One way to achieve this consists in simplifying the layer by dropping the $\text{softmax}$ operator, to obtain

$$\boldsymbol{Y}_{\text{LinAttn}} \coloneqq (\hat{\boldsymbol{Q}}\hat{\boldsymbol{K}}^{\mathsf{T}})\hat{\boldsymbol{V}} = \hat{\boldsymbol{Q}}(\hat{\boldsymbol{K}}^{\mathsf{T}}\hat{\boldsymbol{V}}). \tag{4}$$

This simplification allows one to leverage associativity in matrix multiplication, computing first $\hat{\boldsymbol{K}}^{\mathsf{T}}\hat{\boldsymbol{V}}$ and thus only instantiating much smaller matrices $\hat{\boldsymbol{K}}, \hat{\boldsymbol{Q}}, \hat{\boldsymbol{V}} \in \mathbb{R}^{d \times L}$, rather than the full Attention matrix $\in \mathbb{R}^{L \times L}$.

By comparing (4) with (3), we can draw a direct connection between Linear Attention and (input-dependent) SSMs: indeed, by simplifying $\boldsymbol{\Lambda}_k \equiv \boldsymbol{I}$, one can see that the parameter $\boldsymbol{B}, \boldsymbol{C}, \boldsymbol{X}$ in the SSM mixer cover a similar role as the $\hat{\boldsymbol{K}}, \hat{\boldsymbol{Q}}, \hat{\boldsymbol{V}}$ matrices in Linear Attention. This correspondence is outlined more in detail by Dao & Gu (2024), and further justifies the choice of Mamba in our experiments as a representative for linearized forms of Attention. In this work, we directly leverage such correspondence to ground our distillation recipe, as described in Sec. 3.2.

---

[1]We note that the convolution layer in Mamba applied before the SSM can also perform sequence mixing.

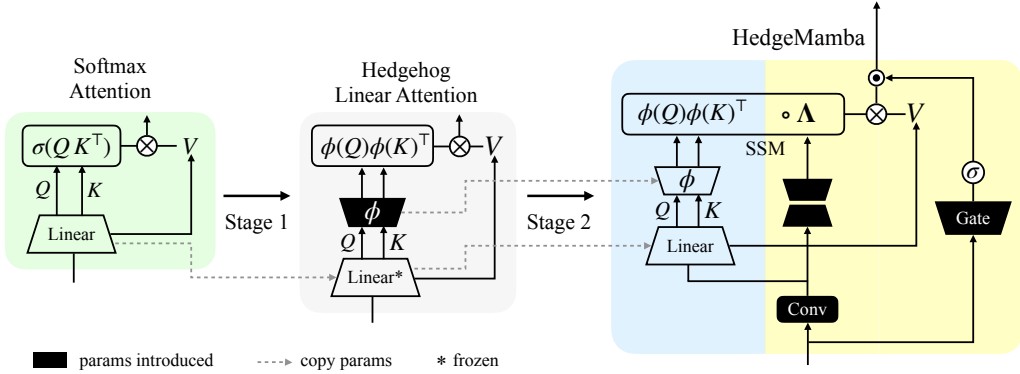

Figure 2: Schematics of the overall approach followed in our two-stage Transformer-to-Mamba distillation recipe. In stage one, we distill vanilla Attention into a linearized version, by learning a feature map $\phi$ which approximates the action of softmax (following the Hedgehog procedure (Zhang et al., 2024)). In stage two, we introduce additional components from the Mamba block, to boost the overall expressivity of the model. The resulting hybrid layer, named HedgeMamba, is then further finetuned to close the performance gap with the original teacher model.

## 3 CROSS-ARCHITECTURE DISTILLATION

In this section, we outline the distillation recipe designed and tested in this project. The driving goal is to leverage the high-level similarity of Transformers and Mamba architectures to improve the distillation procedure. To this end, we split our distillation recipe into two stages. In the first stage, we train a feature map to effectively distill the action of softmax Attention into Linear Attention, following the *Hedgehog* procedure introduced in Zhang et al. (2024). In the second stage, we translate the extracted Linear Attention layer into an initialization for Mamba, leveraging the correspondence outlined in Sec. 2.1 and Dao & Gu (2024), and proceed with further fine-tuning to improve overall performance. We refer to Fig. 2 for an overview of each stage.

### 3.1 STAGE 1: SOFTMAX ATTENTION TO LINEAR ATTENTION

The purpose of our first step is to effectively substitute softmax Attention with a linear variant that can adequately approximate its action. However, as highlighted in Zhang et al. (2024), there is still a notable performance gap between the original softmax Attention and many existing linearizations. Motivated by this, the work in Zhang et al. (2024) focuses instead on distillation via a *learnable feature map*. This is at the core of the *Hedgehog* procedure introduced in Zhang et al. (2024), which we leverage in our distillation method. We briefly define the procedure next.

**Hedgehog** The softmax Attention scores are computed starting from various exponential terms $e^{\boldsymbol{Q}_{l,:}\boldsymbol{K}_{l,:}^{\top}}$, but we want to remove this nonlinearity. Invoking Mercer's theorem (Mercer, 1909) allows us to re-write the positive definite exponential operator as a scalar product of feature vectors,

$$e^{\boldsymbol{x}^{\top}\boldsymbol{x}'} =: \kappa(\boldsymbol{x}, \boldsymbol{x}') = \boldsymbol{\phi}(\boldsymbol{x})^{\top}\boldsymbol{\phi}(\boldsymbol{x}'), \qquad \forall \boldsymbol{x}, \boldsymbol{x}' \in \mathbb{R}^{d}, \tag{5}$$

for a certain feature map $\boldsymbol{\phi}(\boldsymbol{x}) : \mathbb{R}^{d} \to \mathcal{H}$. Specifically, for the exponential kernel (also known as the Gaussian kernel), its feature space $\mathcal{H}$ is infinite-dimensional, and the feature map $\boldsymbol{\phi}(\boldsymbol{x})$ can be approximated via Taylor expansion of $e^{z}$ around $z = 0$. Linear Attention variants that aim to approximate this feature map tend to do so by keeping only the first few terms in the sum in its Taylor expansion (Katharopoulos et al., 2020). However, as pointed out by Zhang et al. (2024), these variants typically do not retain some relevant features of softmax Attention, such as spikiness in the activations and dot-product monotonicity. To overcome this, Zhang et al. (2024) propose to *learn* the feature map in (5) via a (single-layer) MLP:

$$\boldsymbol{\phi}(\boldsymbol{x}) \approx \boldsymbol{\phi}_{\text{MLP}}(\boldsymbol{x}) := \sigma(\boldsymbol{W}\boldsymbol{x} + \boldsymbol{b}), \tag{6}$$

with nonlinearity $\sigma$. The learnable weights $\boldsymbol{W} \in \mathbb{R}^{d \times d}, \boldsymbol{b} \in \mathbb{R}^{d}$ are optimized by **matching the output of each teacher Attention block with that of its Hedgehog-linearized version, via cosine**

**similarity.** In Zhang et al. (2024), the authors showcase how such learnable MLP feature map greatly improves the distillation performance of Linear Attention while remaining computationally efficient (we refer to their work for additional evaluation and implementation details of the Hedgehog procedure). However, while in Zhang et al. (2024) the distillation procedure stops here, in this work we improve this approach by including the learned Hedgehog feature map into the Mamba architecture, and further refining the distillation onto this adapted architecture. More details are outlined next.

### 3.2 STAGE 2: LINEAR ATTENTION TO MAMBA

With the first distillation step, we have identified a way to substitute the $\mathrm{softmax}$ Attention operation in (1) with a Linear Attention one. In the second step, we want to use this Linear Attention solution as an initialization to Mamba, and further fine-tune to improve the distillation performance by leveraging the additional expressivity that the Mamba module provides. In this section we show how to adapt the Mamba layer to achieve exactly this. We refer the resulting Hedgehog-Mamba layer as HedgeMamba.

**Parameters initialization**   As mentioned in Sec. 2, we can match the output of an SSM mixer (2) with that of a Linear Attention layer (4) by substituting $\Lambda_l \equiv I$ and by having the parameters $B, C, X$ cover a similar role as $\hat{K}, \hat{Q}, \hat{V}$. For the specific case of Hedgehog, this translates into the substitutions

$$B(X) \mapsto \hat{K}(X) := \phi_{\mathrm{MLP}}(K(X)), \qquad C(X) \mapsto \hat{Q}(X) := \phi_{\mathrm{MLP}}(Q(X))$$
$$\Lambda \mapsto I, \qquad\qquad \text{and} \qquad X \mapsto \hat{V}(X) := V(X), \tag{7}$$

where $K(X), Q(X), V(X)$ are the key/query/value linear maps from the original $\mathrm{softmax}$ Attention layer (1), while $\phi_{\mathrm{MLP}}$ is the freshly-learned Hedgehog feature map (6). Notice that the original Mamba architecture does not allow for a value transformation $X \mapsto V(X)$ before the application of the SSM mixer, so we modify its implementation to accommodate for this. Moreover, to ensure that the whole Mamba block output matches that of Hedgehog at initialization, we also set the parameters of the gate branch and the convolution so that they reduce to the identity operator. Additional details can be found in App. B.

**Attention scores normalization**   With the substitution in (7), the SSM mixer outputs

$$Y_\phi := \left( \phi_{\mathrm{MLP}}(Q) \phi_{\mathrm{MLP}}(K)^\mathsf{T} \right) V. \tag{8}$$

However, the Attention scores in this formula come in an un-normalized fashion. For the Attention scores formulation to more closely follow the target one in (1), we further include a normalization factor in their definition:

$$Y_\phi \mapsto Y_\phi / \bar{Y}_\phi, \qquad \text{with} \qquad \bar{Y}_\phi := \left( \phi_{\mathrm{MLP}}(Q) \phi_{\mathrm{MLP}}(K)^\mathsf{T} \right) \mathbf{1}. \tag{9}$$

Notice that both $Y_\phi$ and $\bar{Y}_\phi$ can be computed with a single pass through the SSM mixer, provided we expand $V$ with an all-one tensor, and duplicate the state matrix $\Lambda$, that is

$$V \mapsto \mathrm{concat}[V; \mathbf{1}], \qquad \text{and} \qquad \Lambda \mapsto \mathrm{concat}[\Lambda; \Lambda]. \tag{10}$$

**Fine-tuning**   With Mamba initialized as in (7), and modified to accommodate for normalization as per (9) and (10), we are ready to resume training and enter the second stage of our distillation procedure. This amounts to **fine-tuning the whole architecture** (except the embedding layers) **via Cross-Entropy loss with respect to the ground-truth**. Particularly, we also unlock the additional convolution and gate branches available in the original Mamba block, completing our definition of the HedgeMamba layer: see also Fig. 2 for an outline of its components fine-tuned in this final stage.

The key argument from our work is that by equipping Mamba with a Hedgehog initialization we can recover an overall better recipe for cross-architectural distillation. As we illustrated in this section, our two-stage approach is theoretically grounded in Mercer's theorem, together with the superior expressive power of Mamba over vanilla Linear Attention. In the following section, we proceed to justify our approach also empirically, by benchmarking models trained following our recipe.

## 4 RESULTS

In this section, we present an extensive evaluation of the distillation procedure outlined in Sec. 3. Specifically, we conduct ablation, scaling, and sensitivity studies with Pythia-1B teacher model and 10B distillation tokens across several key axes: (i) mixer architecture, by systematically expanding vanilla Hedgehog Linear Attention with components from Mamba (Tab. 2); (ii) sensitivity analysis on token budget allocation in different stages (Tab. 3); and (iii) scaling with respect to number of distillation tokens (Tab. 4). Our default settings are highlighted in their respective tables. In App. A.1 we further expand on the results in this section, including results on applying our distillation procedure to different model sizes (160M, 410M, and 1B), and reporting standard error for results in Tab. 2 to 4.

**Experimental setup** For all experiments we use standard Pythia-1B (Biderman et al., 2023) as teacher model. This model has been widely adopted by the open source community, and the suite provides both the model weights (for different scales), and the detailed full training procedure. We distill our models using the OpenWebText (Gokaslan et al., 2019) dataset. This is an open-source reproduction of the dataset used to train GPT2 (Radford et al., 2019), and it is commonly used in language modeling research (Biderman et al., 2023; Sanh et al., 2019; Dao et al., 2022; Shoeybi et al., 2019; Zhuang et al., 2021). We employ the same GPT-NeoX tokenizer used for the original Pythia and Mamba models, making our results directly comparable. This amounts to a total of about 9B tokens available for training. We keep a $0.0005\%$ split on the dataset for validation, which corresponds to 4M tokens, as in prior works (Dao et al., 2022). Unless otherwise reported, we use a total of 10B tokens (roughly corresponding to 1.1 epochs of OpenWebText) which, to the best of our knowledge, establishes our work as the currently largest sensitivity study with respect to token budget in distillation. We evaluate the final distilled student models both in terms of upstream perplexity as well as performance on selected downstream tasks. For the latter, we rely on the `lm-eval-harness` (Gao et al., 2021) test suite, and consider language understanding and common sense reasoning used in prior work (Biderman et al., 2023; Gu & Dao, 2023; Dao & Gu, 2024). More specifically, we report accuracy scores for ARC-Easy (Clark et al., 2018), Social IQA Sap et al. (2019), PiQA Bisk et al. (2020), Lambada (Paperno et al., 2016), BoolQ (Clark et al., 2019), RACE (Lai et al., 2017), LogiQA (Liu et al., 2020), and WinoGrande (Sakaguchi et al., 2019), and accuracy normalized by sequence length for ARC-Challenge (Clark et al., 2018) and HellaSwag (Zellers et al., 2019), as in (Bick et al., 2024; Gu & Dao, 2023; Dao & Gu, 2024; Sanh et al., 2019). We refer the reader to (Gao et al., 2021) for more details regarding evaluation.

**Training** In the first stage of recipe (Sec. 3.1), we replace the Attention block in the teacher model with the Hedgehog linearization, with the goal of learning the feature map (6). Except for the parameters defining this feature map (which are learned from scratch in stage 1), all the other parameters are copied directly from the teacher model and kept frozen. These include MLPs, layer norms, and input-output embedding matrices. We match the output of each Transformer layer (consisting of MLP, sequence mixer, and residual stream) in the student model with those from the teacher via cosine embedding matching loss. We use 1B tokens for stage 1 of our distillation recipe with batch size 48 and sequence length 1024, which corresponds to 20K training steps. In the second stage (Sec. 3.2), we introduce additional Mamba parameters, initialized to the identity operator (see App. B). We keep the input-output embedding layers frozen, and finetune the rest of the model with standard cross-entropy loss. Second-stage training continues for another 9B tokens, corresponding to additional 180K training steps.

**Implementation remarks** For our implementation of the HedgeMamba layer in Fig. 2, we directly adapt the Mamba code, while still leveraging their hardware-aware CUDA selective scan, as to not sacrifice efficiency[2] (see the corresponding code in App. C). We use the teacher models implementations and pretrained weights directly from the HuggingFace Transformers library (Wolf et al., 2020). Student models are implemented by swapping the softmax Attention modules from the teacher with Mamba Mixer modules from Gu & Dao (2023), equipped with the Hedgehog feature maps from Zhang et al. (2024). Further implementation details are in App. A.2.

---

[2]We point out that Mamba selective scan implementation, albeit perfectly parallel, imposes a hard-cap of 256 on model dimension (pprp, 2024), forcing serialization for larger values. In our experiments we reach 2048, resulting in inflated figures ($> 8\times$) for our training times (around 12d 9h on a 8xA100 node to distill 10B tokens using a 1B model). We refer then to distillation token budget as a more reliable metric for our procedure cost.

**Baseline comparison**  Our main objective is to showcase the ability of our two-stage recipe to retain the performance of the original teacher model. We point out that this is not guaranteed, given the substantial differences with the teacher's architecture, and the fact that we consider a pure Linear Attention variant—and not hybrids like in some prior works (Wang et al., 2024; Bick et al., 2024). Moreover, the original Pythia-1B teacher model was trained with 300B tokens (Biderman et al., 2023), but we distill it with **only 10B, corresponding to ∼2.7% of the token budget used to train the teacher** (334B tokens). The distillation results are reported in Tab. 1. We compare our recipe against the Hedgehog baseline, distilled using the same cosine matching objective in stage 1 and cross-entropy in stage 2, although in this case the split in distillation tokens is 50/50 among the two stages, as per their original work (Zhang et al., 2024; 2025). Overall, our approach manages to retain the teacher performance for the large part, scoring a perplexity of 14.11 versus the original 13.86. We also point out that our approach outperforms the scaled-up Hedgehog baseline with 10B tokens in both upstream and downstream performance, highlighting the efficacy of our approach. As additional baseline (not reported here), we tested against naïve direct distillation onto a Mamba architecture, but consistently resulted in unsatisfactory results (PPL>100), confirming the findings from Bick et al. (2024).

Table 1: Comparison with teacher and prior work.

| Model (1B) | ↓ PPL | ↑ Arc-C | Arc-E | SIQA | PiQA | Lambada | BoolQ | RACE | LogiQA | WinoG | HSwag |
|---|---|---|---|---|---|---|---|---|---|---|---|
| Pythia (Teacher) | 13.86 | 27.04 | 56.98 | 39.86 | 70.72 | 42.07 | 60.82 | 32.92 | 22.12 | 53.43 | 47.16 |
| Hedgehog (Baseline) | 14.89 | 26.45 | 52.74 | 38.38 | 68.01 | 30.60 | 54.80 | 30.43 | 21.65 | 50.91 | 40.79 |
| HedgeMamba (Ours) | **14.11** | 27.13 | 53.66 | 39.76 | 68.72 | 32.31 | 55.20 | 30.91 | 20.89 | 52.17 | 41.87 |

**Ablating Mamba components**  In the ablation study in Tab. 2, we investigate the role of the additional Mamba components included in stage 2 in improving the final performance of the student model. Specifically, we consider simple Hedgehog as a baseline (Zhang et al., 2024), and systematically add the following components from Mamba: (+SSM) the SSM mixer parameters, particularly the learnable causal mask $\Lambda$ and input and output matrices $C$ and $B$ from (2); (+Conv) the short-convolution layer at the input; (+Gate) the gate branch with SiLU non-linearity. These added components are initialized to behave like the identity, not to affect the Hedgehog feature map learned in stage 1 (see App. B). The other SSM mixer parameters $C$ and $B$ are instead directly copied from their equivalent in the Hedgehog module, as described in (7). To analyze the impact of the new introduced mixer components in a targeted manner, all the other ablation parameters are kept fixed: particularly, we use 10B distillation tokens, split evenly (50/50) among stages 1 and 2. The corresponding results are reported in Tab. 2. There, we can see how each additional Mamba component contributes to improving the performance of vanilla Hedgehog. Interestingly, the largest improvements in perplexity and average downstream performance are yielded by the gate branch. This finding is consistent with recent works (Qiu et al., 2025; Hua et al., 2022; Bondarenko et al., 2023), that suggest adding a gate branch to Attention modules to improve their performance.

Table 2: Mixer architecture ablations: perplexity on validation set and downstream task performance.

| Model (1B) | #params | ↓ PPL | ↑ Arc-C | Arc-E | SIQA | PiQA | Lambada | BoolQ | RACE | LogiQA | WinoG | HSwag |
|---|---|---|---|---|---|---|---|---|---|---|---|---|
| Hedgehog | 1,014M | 14.89 | 26.45 | 52.74 | 38.38 | 68.01 | 30.60 | 54.80 | 30.43 | 21.66 | 50.91 | 40.79 |
| +SSM | 1,020M | 14.89 | 26.54 | 52.90 | 38.02 | 68.23 | 31.24 | 55.63 | 30.05 | 22.73 | 51.38 | 40.77 |
| +Conv | 1,020M | 14.89 | 26.62 | 52.74 | 38.28 | 68.93 | 31.63 | 55.84 | 30.14 | 22.43 | 51.78 | 40.74 |
| +Gate(HedgeMamba) | 1,087M | **14.58** | 26.19 | 53.11 | 39.56 | 68.77 | 32.16 | 57.61 | 31.00 | 24.42 | 50.99 | 41.81 |

**Token budget allocation between stages**  One relevant design choice for our recipe consists in determining how to best allocate our distillation tokens budget among the two stages. In Tab. 3 we verify this empirically, and report student performance evaluations when varying how the distillation tokens are split between stages 1 and 2. Notice that in the original Hedgehog paper Zhang et al. (2024), the authors settle for a 50/50 split; our results instead show that it is progressively more beneficial to invest into stage 2, up to 90% of the total token budget. Still, both stages are needed to guarantee performance, as testified by the poor results attained by the extreme 100/0 and 0/100

Table 3: Sensitivity analysis on the token budget allocation. Total 10B distillation tokens. Distillation on full HedgeMamba, with convolution layer and gate branch. Notice that a 100/0 split indicates that no fine-tuning is performed, and all tokens are used for learning the Hedgehog map in stage 1 (S1); conversely, 0/100 indicates only fine-tuning (S2) on the HedgeMamba architecture.

| | Tokens split | | ↑ Downstream evals | | | | | | | | | |
|---|---|---|---|---|---|---|---|---|---|---|---|---|
| | S1 / S2 (%) | ↓ PPL | Arc-C | Arc-E | SIQA | PiQA | Lambada | BoolQ | RACE | LogiQA | WinoG | HSwag |
| Hedgehog (no FT) | 100 / 0 | 25.71 | 25.85 | 48.70 | 36.34 | 66.49 | 12.12 | 61.47 | 27.27 | 20.58 | 50.83 | 26.14 |
| | 90 / 10 | 16.15 | 25.00 | 52.06 | 38.69 | 68.93 | 28.08 | 56.15 | 30.24 | 22.43 | 51.14 | 39.69 |
| | 75 / 25 | 15.18 | 26.71 | 52.31 | 38.59 | 69.26 | 30.66 | 60.61 | 30.24 | 20.58 | 49.96 | 41.02 |
| | 50 / 50 | 14.58 | 26.19 | 53.11 | 39.56 | 68.77 | 32.16 | 57.61 | 31.00 | 24.42 | 50.99 | 41.81 |
| | 25 / 75 | 14.25 | 26.19 | 53.91 | 39.71 | 68.93 | 31.90 | 55.41 | 30.81 | 21.35 | 51.30 | 41.59 |
| Default | 10 / 90 | **14.11** | 27.13 | 53.66 | 39.76 | 68.72 | 32.31 | 55.20 | 30.91 | 20.89 | 52.17 | 41.87 |
| Finetune only | 0 / 100 | 17.08 | 26.11 | 50.67 | 37.31 | 67.03 | 27.61 | 54.01 | 30.33 | 21.35 | 50.51 | 40.25 |

splits. We point out however that the second stage is generally more computationally expensive: total training time[2] increases from 12d 9h to 13d 16h for the 0/100 split.

**Scaling number of distillation tokens**    For reference, in Tab. 4 we report how the final performance of our student model scales with respect to the number of distillation tokens available. We distill Pythia-1B onto full HedgeMamba, with convolution layer and gate branch. The tokens budget split between the two stages is fixed at the optimal 10/90, and we only vary total budget. Overall, student perplexity improves as the token budget increases, and at 10B it has not yet reached saturation.

Table 4: Scaling study on the distillation token budget.

| Token budget | ↓ PPL | ↑ Arc-C | Arc-E | SIQA | PiQA | Lambada | BoolQ | RACE | LogiQA | WinoG | HSwag |
|---|---|---|---|---|---|---|---|---|---|---|---|
| 1B | 16.56 | 26.19 | 52.27 | 38.74 | 67.68 | 27.32 | 57.49 | 29.76 | 20.43 | 52.25 | 40.67 |
| 2B | 15.61 | 25.94 | 51.05 | 38.79 | 69.04 | 29.30 | 56.45 | 29.57 | 23.04 | 51.85 | 40.29 |
| 3B | 15.15 | 25.09 | 52.69 | 38.43 | 69.10 | 30.56 | 56.57 | 29.28 | 23.04 | 51.93 | 41.03 |
| 10B | **14.11** | 27.13 | 53.66 | 39.76 | 68.72 | 32.31 | 55.20 | 30.91 | 20.89 | 52.17 | 41.87 |

# 5    CONCLUSION

In this paper, we propose a novel recipe to distill a Transformer model into an SSM-based architecture. The goal is to allow the user to reduce inference time (from quadratic in sequence length, as in classical softmax Attention, to linear), without having to train a new architecture from scratch, and instead aptly leveraging already-available pretrained models. The design of our architecture relies on a principled approach, first approximating softmax Attention with a Linear Attention variant, and then use it to initialize a Mamba block, to boost its expressivity. Mirroring these steps, the distillation procedure is also composed of two stages: the first with the goal of aligning Attention weights, and the second allowing for further fine-tuning of the whole architecture. In particular, the inclusion of this first stage has proven to boost outputs alignment between student and teacher, over naïve direct distillation. The effectiveness of the procedure is evaluated both in terms of perplexity and performance on downstream tasks, showcasing its overall ability to preserve the teacher performance.

**Limitations and future work**    To maintain a clear focus, we targeted our analysis on a specific Transformer architecture (namely, Pythia). In principle, our recipe is flexible enough to be extended to other Attention-based models, but we have not investigated its effectiveness on other variants, also in light of the computational resources generally required for distillation (see Sec. 4). For similar reasons, we do not isolate the effect of distillation dataset quality on the final student performance, and limit our experiments to OpenWebText only. Finally, in the scope of this work, we investigated *one* way of boosting the student model expressivity, that is by incorporating components from the Mamba architecture: the space of possible extensions, however, remains open to additional exploration which might further increase final performance. This notwithstanding, we believe that our work represents a meaningful step in this exploration, covering a previously unexplored approach to bridging the gap between Attention and Mamba.

## REPRODUCIBILITY STATEMENT

To ensure the reproducibility of this work, we train our models on publicly available data using open-source libraries and we will publish the code. We also provide comprehensive details on our methodology and implementation. Specifically, the full architectural schematics of HedgeMamba and its components are presented in Fig. 4, with detailed pseudocode provided in App. C. We also specify the initialization strategies for Mamba parameters in App. B.2. Our training setup, including hardware, optimizer, learning rate schedules, and regularization techniques, is thoroughly documented in App. A.2. Finally, evaluation metrics and their associated standard errors, derived from 100,000 bootstrap repetitions using lm-eval settings, are reported in Tab. 5.

## ETHICS STATEMENT

We adhere to ICLR's Code of Ethics. In this work we introduce an improved recipe to distill Transformers into Mamba architectures. Our method can help reducing the inference cost of generative models, reducing energy usage, and helping to democratize the access to AI capabilities for researchers and organizations with limited resources. Like any other language modeling research, the models derived from this work could potentially be misused.

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

## A  ADDITIONAL RESULTS AND IMPLEMENTATION DETAILS

### A.1  EXPANDED RESULTS

**Summary**

In this section we expand the results in Sec. 4 and find that:

- Increasing the scale of the architectures (from 160M, to 410M and 1B) reduces PPL and increases all other performances, as shown in Tab. 6.
- Even though stage 2 provides the larger benefits in decreasing overall PPL, the role of stage 1 is key in improving performance, as shown in Fig. 3.

In this section we expand upon the results provided in Sec. 4. Particularly, Tab. 5 collects the results in Tab. 2 to 4, and reports also their associated standard error over 100,000 bootstrap repetitions (as per default `lm-eval` settings[3]). Overall, standard deviation remains low across all tasks, indicating their robustness.

Table 5: Results from experiments in Sec. 4, including standard error for LM evaluations.

| Model (1B) | $\uparrow$Arc-C | Arc-E | SIQA | PiQA | Lambada | BoolQ | RACE | LogiQA | WinoG | HSwag |
|---|---|---|---|---|---|---|---|---|---|---|
| Pythia(Teacher) | $27.05_{1.30}$ | $56.99_{1.02}$ | $39.87_{1.11}$ | $70.73_{1.06}$ | $42.07_{0.69}$ | $60.82_{0.85}$ | $32.92_{1.45}$ | $22.12_{1.62}$ | $53.43_{1.40}$ | $47.16_{0.50}$ |
| *Architecture ablations* | | | | | | | | | | |
| Hedgehog baseline | $26.45_{1.29}$ | $52.74_{1.02}$ | $38.38_{1.10}$ | $68.01_{1.09}$ | $30.60_{0.64}$ | $54.80_{0.87}$ | $30.43_{1.42}$ | $21.66_{1.62}$ | $50.91_{1.41}$ | $40.79_{0.49}$ |
| +SSM | $26.54_{1.29}$ | $52.90_{1.02}$ | $38.02_{1.10}$ | $68.23_{1.09}$ | $31.24_{0.65}$ | $55.63_{0.87}$ | $30.05_{1.42}$ | $22.73_{1.64}$ | $51.38_{1.40}$ | $40.77_{0.49}$ |
| +Conv | $26.62_{1.29}$ | $52.74_{1.02}$ | $38.28_{1.10}$ | $68.93_{1.08}$ | $31.63_{0.65}$ | $55.84_{0.87}$ | $30.14_{1.42}$ | $22.43_{1.64}$ | $51.78_{1.40}$ | $40.74_{0.49}$ |
| +Gate(HedgeMamba) | $26.19_{1.28}$ | $53.11_{1.02}$ | $39.56_{1.11}$ | $68.77_{1.08}$ | $32.16_{0.65}$ | $57.61_{0.86}$ | $31.00_{1.43}$ | $24.42_{1.69}$ | $50.99_{1.40}$ | $41.81_{0.49}$ |
| *Sensitivity: token allocation – stage1 / stage2 (%) split* | | | | | | | | | | |
| 100 / 0 | $25.85_{1.28}$ | $48.70_{1.03}$ | $36.34_{1.09}$ | $66.49_{1.10}$ | $12.12_{0.45}$ | $61.47_{0.85}$ | $27.27_{1.38}$ | $20.58_{1.60}$ | $50.83_{1.40}$ | $26.14_{0.48}$ |
| 90 / 10 | $25.00_{1.27}$ | $52.06_{1.03}$ | $38.69_{1.10}$ | $68.93_{1.08}$ | $28.08_{0.65}$ | $56.15_{0.87}$ | $30.24_{1.43}$ | $22.43_{1.59}$ | $51.14_{1.40}$ | $39.69_{0.49}$ |
| 75 / 25 | $26.71_{1.29}$ | $52.31_{1.02}$ | $38.59_{1.11}$ | $69.26_{1.08}$ | $30.66_{0.65}$ | $60.61_{0.87}$ | $30.24_{1.43}$ | $20.58_{1.61}$ | $49.96_{1.40}$ | $41.02_{0.49}$ |
| 50 / 50 | $26.19_{1.28}$ | $53.11_{1.02}$ | $39.56_{1.11}$ | $68.77_{1.08}$ | $32.16_{0.65}$ | $57.61_{0.86}$ | $31.00_{1.43}$ | $24.42_{1.69}$ | $50.99_{1.40}$ | $41.81_{0.49}$ |
| 25 / 75 | $26.19_{1.28}$ | $53.91_{1.02}$ | $39.71_{1.11}$ | $68.93_{1.08}$ | $31.90_{0.64}$ | $55.41_{0.85}$ | $30.81_{1.42}$ | $21.35_{1.60}$ | $51.30_{1.41}$ | $41.59_{0.49}$ |
| 10 / 90 | $27.13_{1.30}$ | $53.66_{1.02}$ | $39.76_{1.11}$ | $68.72_{1.08}$ | $32.31_{0.63}$ | $55.20_{0.87}$ | $30.91_{1.42}$ | $20.89_{1.64}$ | $52.17_{1.40}$ | $41.87_{0.49}$ |
| 0 / 100 | $26.11_{1.28}$ | $50.67_{1.03}$ | $37.31_{1.09}$ | $67.03_{1.10}$ | $27.61_{0.62}$ | $54.01_{0.87}$ | $30.33_{1.42}$ | $21.35_{1.61}$ | $50.51_{1.41}$ | $40.25_{0.49}$ |
| *Scaling: overall token budget* | | | | | | | | | | |
| 1B | $26.19_{1.28}$ | $52.27_{1.03}$ | $38.74_{1.10}$ | $67.68_{1.09}$ | $27.32_{0.61}$ | $57.49_{0.87}$ | $29.76_{1.41}$ | $20.43_{1.58}$ | $52.25_{1.40}$ | $40.67_{0.49}$ |
| 2B | $25.94_{1.28}$ | $51.05_{1.03}$ | $38.79_{1.10}$ | $69.04_{1.08}$ | $29.30_{0.63}$ | $56.45_{0.87}$ | $29.57_{1.41}$ | $23.04_{1.65}$ | $51.85_{1.40}$ | $40.29_{0.49}$ |
| 3B | $25.09_{1.27}$ | $52.69_{1.02}$ | $38.43_{1.10}$ | $69.10_{1.08}$ | $30.56_{0.64}$ | $56.57_{0.87}$ | $29.28_{1.41}$ | $23.04_{1.65}$ | $51.93_{1.40}$ | $41.03_{0.49}$ |
| 10 | $27.13_{1.30}$ | $53.66_{1.02}$ | $39.76_{1.11}$ | $68.72_{1.08}$ | $32.31_{0.63}$ | $55.20_{0.87}$ | $30.91_{1.42}$ | $20.89_{1.64}$ | $52.17_{1.40}$ | $41.87_{0.49}$ |

In Tab. 6 we report an additional analysis on the performance of our distillation procedure when applied to models of various sizes (160M and 410M, on top of our already-presented 1B results). Overall, the results confirms the trend of HedgeMamba consistently showing improvements over the Hedgehog approach.

Figure 3 provides the detailed evolution of the validation perplexity during training, for the token allocation splits discussed in Tab. 3. As a reminder, we train for a total 200K steps: each train step uses 49,200 tokens, so that the total number of tokens used in training are 49,200×200,000=9,840,000,000 ≈ 10B. The training steps are allocated between the first and second stage of our recipe in Sec. 3 depending on the split considered: for instance, a 10/90 split signifies that 10% of the training steps (i.e. 20K steps) are used for stage 1 and 90% (180K steps) for stage 2. From Fig. 3 we can infer that, even though stage 2 provides the larger benefits in decreasing overall PPL, the role of stage 1 is key in improving performance. Allocating all training tokens to stage 2, in fact, causes PPL to stall at a much higher value than if we allocated even a small fraction (with as small as 10% giving the best results) also to stage 1.

---

[3]LM harness evaluation estimates the Standard Error of the Mean (SEM) using bootstrap resampling: it repeatedly samples a set of multiple choice questions (for a specified number of bootstrap iterations, 100K in our case) and then calculates the SEM of the metric scores obtained from these samples. Relevant code for this error computation can be found in https://github.com/EleutherAI/lm-evaluation-harness.

Table 6: Scaling analysis with respect to model size.

| | Model | ↓ PPL | ↑ Arc-C | Arc-E | SIQA | PiQA | Lambada | BoolQ | RACE | LogiQA | WinoG | HSwag |
|---|---|---|---|---|---|---|---|---|---|---|---|---|
| 160M | Pythia(Teacher) | 39.38 | 23.63 | 43.64 | 36.75 | 62.30 | 22.38 | 56.88 | 28.71 | 19.05 | 51.22 | 30.28 |
| | Hedgehog (Baseline) | 35.95 | 18.26 | 42.47 | 37.05 | 61.15 | 14.48 | 59.66 | 26.41 | 21.04 | 50.43 | 28.93 |
| | HedgeMamba (Ours) | 26.84 | 23.04 | 43.27 | 37.36 | 60.88 | 16.34 | 57.68 | 26.03 | 19.35 | 51.07 | 29.71 |
| 410M | Pythia(Teacher) | 16.50 | 24.32 | 51.89 | 38.95 | 66.70 | 36.60 | 60.58 | 30.72 | 21.97 | 53.27 | 40.62 |
| | Hedgehog (Baseline) | 17.66 | 19.97 | 47.31 | 37.97 | 65.23 | 24.04 | 48.44 | 28.04 | 19.35 | 50.28 | 34.63 |
| | HedgeMamba (Ours) | 16.48 | 23.81 | 49.54 | 38.69 | 64.69 | 25.91 | 51.68 | 28.42 | 21.35 | 52.80 | 36.28 |
| 1B | Pythia (Teacher) | 13.86 | 27.04 | 56.98 | 39.86 | 70.72 | 42.07 | 60.82 | 32.92 | 22.12 | 53.43 | 47.16 |
| | Hedgehog (Baseline) | 14.89 | 26.45 | 52.74 | 38.38 | 68.01 | 30.60 | 54.80 | 30.43 | 21.65 | 50.91 | 40.79 |
| | HedgeMamba (Ours) | 14.11 | 27.13 | 53.66 | 39.76 | 68.72 | 32.31 | 55.20 | 30.91 | 20.89 | 52.17 | 41.87 |

## A.2 IMPLEMENTATION DETAILS

We use PyTorch with Distributed Data Parallel with mixed precision (bfloat16) for training. For our implementation of the HedgeMamba layer in Fig. 2, we directly adapt the Mamba code, while still leveraging their hardware-aware CUDA selective scan, as to not sacrifice efficiency. We point out that Mamba selective scan implementation, albeit perfectly parallel, imposes a hard-cap of 256 on model dimension (pprp, 2024), forcing serialization for larger values. In our experiments we reach 2048, resulting in inflated figures ($> 8\times$) for our training times (around 12d 9h on a 8xA100 node to distill 10B tokens using a 1B model). We refer then to distillation token budget as a more reliable metric for our procedure cost (see the corresponding code in App. C).

We use the teacher models implementations and pretrained weights directly from the HuggingFace Transformers library (Wolf et al., 2020). Student models are implemented by swapping the softmax Attention modules from the teacher with Mamba Mixer modules from Gu & Dao (2023), equipped with the Hedgehog feature maps from Zhang et al. (2024).

All the models are distilled on a compute node with 8 NVIDIA A100 GPUs. We use AdamW optimizer ($\beta_1 = 0.9$, $\beta_2 = 0.95$) with linear warm-up and cosine decay to $0.1\times$ peak LR schedule in our distillation procedure. Empirically, for models of size 1B, a peak learning rate of 0.01 was found

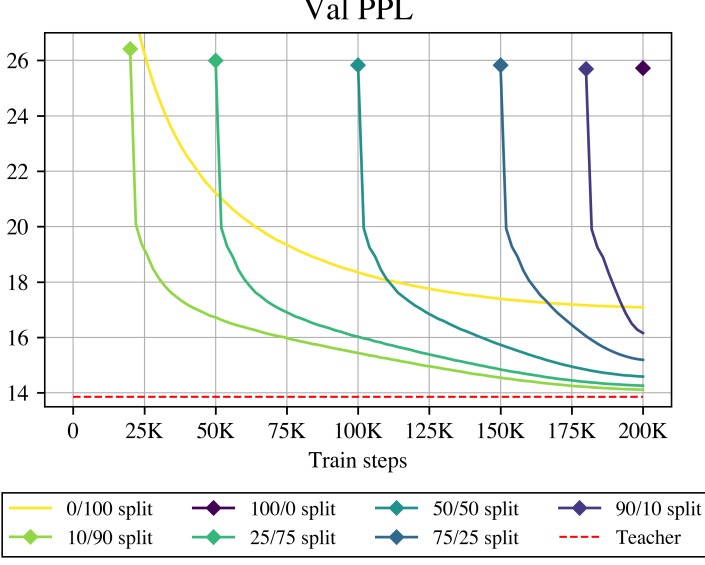

Figure 3: Visualizing evolution of validation perplexity during stage 2 training, for different stage1/stage2 token allocation splits. ♦ indicates the PPL value at completion of stage 1. All sensitivity studies were run for 200K training steps, corresponding to roughly 10B tokens.

to be suitable for stage 1, while a learning rate in the 1e-5 range worked best for stage 2. Following prior work (Gu & Dao, 2023; Dao & Gu, 2024), we use gradient clipping of 1.0 and weight decay 0.1.

# B ADDITIONAL ARCHITECTURE DETAILS

## B.1 FULL ARCHITECTURES SCHEMATICS

For reference, the diagrams in Fig. 4 describe the complete architectures discussed in this project. The Pythia Transformer Biderman et al. (2023), which is the teacher model used throughout our experiments, appears on the top. The original Mamba architecture Gu & Dao (2023) is reported on the bottom right. Notice how in both architectures the main components are a sequence mixer (Attention for the Transformer, and the SSM Mixer for Mamba) interwoven with MLPs (in Mamba, this role is covered by the gate branch). In the bottom left, we can see how the Hedgehog block attains Attention linearization. Finally, in the bottom-middle of Fig. 4, acting as a bridge between the Hedgehog and Mamba architectures, we illustrate the HedgeMamba hybrid we proposed and used as student in this work: most of the architecture is inherited directly from Pythia, but the sequence mixer is substituted with a combination of Hedgehog and components from Mamba.

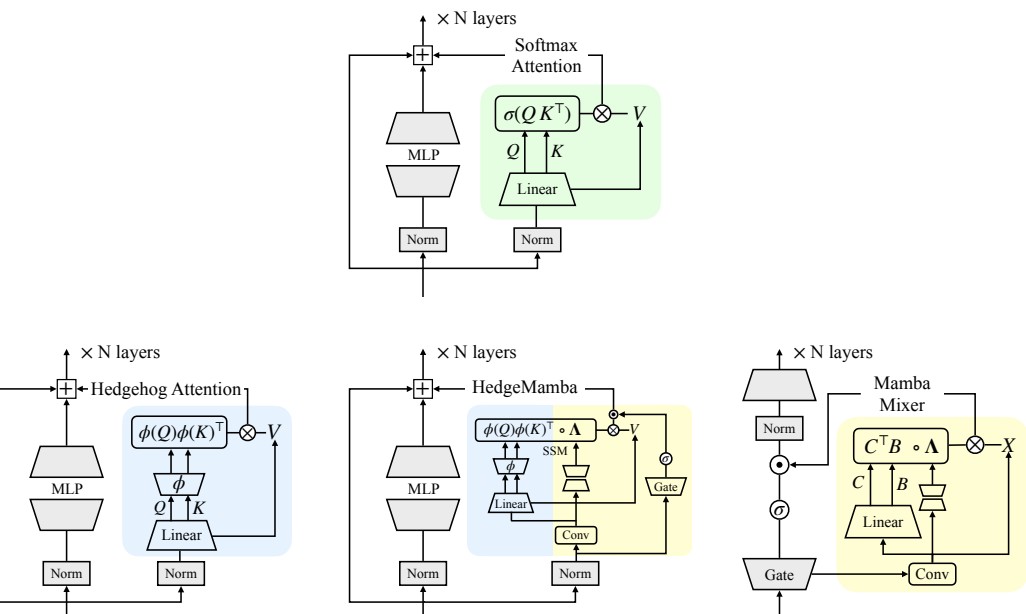

Figure 4: Schematics of architectures discussed in this project. Top: Pythia Transformer. Bottom, from left to right: Hedgehog, HedgeMamba, and Mamba.

## B.2 INITIALIZATION OF MAMBA PARAMETERS IN STAGE 2

A distinguishing feature of Mamba consists in its ability to prescribe a learnable causal mask Dao & Gu (2024) via its state matrix $\mathbf{\Lambda}$ (3). On top of this, the Mamba block also contains a short sequence-wise convolution and a gate branch. All these features contribute to set Mamba apart from other Linear Attention alternatives. We only leverage these three additional components in the second stage of our distillation method, while in the first stage we freeze them and make sure that they have no effect on the block output (see also the schematics in Fig. 2). This can be achieved by an opportune parameter initialization, described next.

**State matrix** The state matrix $\mathbf{\Lambda}$ in Mamba is obtained by exponentiating a product of two parameters: a rate-of-decay $\lambda \in \mathbb{R}^{N \times d}$ and a time-step $\Delta = \Delta(\boldsymbol{X}) \in \mathbb{R}^N$. In particular, the latter is recovered by applying an MLP to the input $\boldsymbol{X} \in \mathbb{R}^{L \times d}$: two linear applications with weights

and biases $(\boldsymbol{W}_d, \boldsymbol{b}_d) \in \mathbb{R}^{d \times d_r} \times \mathbb{R}^{d_r}$ and $(\boldsymbol{W}_u, \boldsymbol{b}_u) \in \mathbb{R}^{d_r \times N} \times \mathbb{R}^N$, respectively, followed by a `SoftPlus` nonlinearity. The overall formula is given by

$$\boldsymbol{\Lambda}_l = e^{-\lambda \odot (\Delta_l \otimes \mathbf{1}^\top)}, \quad \text{with} \quad \Delta_l = \texttt{SoftPlus}((\boldsymbol{X}_{l,:}\boldsymbol{W}_d + \boldsymbol{b}_d)\boldsymbol{W}_u + \boldsymbol{b}_u), \qquad \forall l = 1 \ldots L. \quad (11)$$

We reduce the operation to identity by imposing $\lambda \equiv \mathbf{0}$. Notice that $\Delta_l$ also affects the definition of $\boldsymbol{B}$ in (3): to nullify its effect, we must have $\Delta_l \equiv \mathbf{1}$. To this end, we also impose $\boldsymbol{W}_u \equiv \mathbf{0}$ and $\boldsymbol{b}_u \equiv \texttt{SoftPlus}^{-1}(1) \cdot \mathbf{1} \approx 0.541324 \cdot \mathbf{1}$.

**Convolution** The convolution component applies the following operation: given an input $\boldsymbol{X} \in \mathbb{R}^{L \times d}$, and its kernel weights $\boldsymbol{W} \in \mathbb{R}^{\kappa \times d}$ (for a kernel of size $\kappa$) and biases $\boldsymbol{b} \in \mathbb{R}^d$, the output is given by:

$$\boldsymbol{Y}_{l,:} = \boldsymbol{b} + \sum_{i=1}^{\kappa} \boldsymbol{W}_{i,:} \odot \boldsymbol{X}_{l-\kappa+i,:}. \quad (12)$$

To collapse this to the identity operation, it suffices then to pick $\boldsymbol{b} \equiv \mathbf{0}$, $\boldsymbol{W}_{\kappa,:} \equiv \mathbf{1}$, and $\boldsymbol{W}_{i \neq \kappa,:} \equiv \mathbf{0}$. Notice that in the original Mamba block, the convolution is followed by a nonlinearity, which acts before the SSM mixing layer. In our architecture, we remove this nonlinearity, on the ground that it is already subsumed by the Hedgehog MLP (6).

**Gate** The gate branch, on the other hand, consists of a linear layer (of weights $\boldsymbol{W} \in \mathbb{R}^{d \times d}$ and biases $\boldsymbol{b} \in \mathbb{R}^d$), followed by a `SiLU` nonlinearity. The output is then element-wise multiplied by the output of the SSM mixer (here denoted as $\boldsymbol{X}_{SSM}$). Overall, this amounts to

$$\boldsymbol{Y} = \boldsymbol{X}_{SSM} \odot \texttt{SiLU}(\boldsymbol{X}\boldsymbol{W} + \boldsymbol{b}). \quad (13)$$

To obtain the identity, then, it suffices to set $\boldsymbol{W} \equiv \mathbf{0}$, and $\boldsymbol{b} = \texttt{SiLU}^{-1}(1) \cdot \mathbf{1} \approx 1.27846 \cdot \mathbf{1}$.

## C  PSEUDOCODE

Here we provide pseudocode for the application of the forward pass of the student model used in our experiments.

More in detail, Lst. 1 reports the implementation of the whole adapted Pythia block Biderman et al. (2023). As in the original Pythia model, the flow of operations in this block is split into two branches (see also Fig. 4). On the one hand, we have an MLP with pre-normalization; on the other, the vanilla Attention layer is substituted with the hybrid HedgeMamba layer described in Sec. 3.2.

```python
# --- modified pythia layer powered by hedge-mamba module --- #
class HedgeMambaLayer(GPTNeoXLayer):
    def __init__(
        self,
        config: PretrainedConfig,
    ):
        super().__init__(config)  # standard pythia layer init
        self.mixer_config = mixer_config
        # overwrite attention module with mamba-mixer
        self.attention = HedgeMambaMixer(config)

    def forward(
        self,
        hidden_states: torch.FloatTensor,
        cache_params: Optional[MambaCache] = None,
    ):
        # attention stream
        attn_output = self.input_layernorm(hidden_states)
        attn_output = self.attention(attn_output, cache_params=cache_params)
        attn_output = self.post_attention_dropout(attn_output)

        # mlp stream
        mlp_output = self.mlp(self.post_attention_layernorm(hidden_states))
        mlp_output = self.post_mlp_dropout(mlp_output)

        # pythia layer with parallel MLP and attention streams
        # pseudocode: x = x + attn(ln1(x)) + mlp(ln2(x))
        hidden_states = hidden_states + attn_output + mlp_output
        return hidden_states
```

Listing 1: Implementation of our `PyMambaLayer`, which replaces the Softmax Attention module in the Pythia Transformer with our `HedgeMamba` mixer. Code for the latter is provided in Lst. 2.

HedgeMamba is the core module introduced in our work, and pseudocode for its implementation is detailed in Lst. 2. Its code blueprint closely follows the one for the Mamba SSM mixer Gu & Dao (2023), including a gate branch and a short convolution before the mixer application, but presents three main differences: (i) the SSM parameters $B, C$ (covering the roles of *keys* and *queries* in Linear Attention) are further modified according to the Hedgehog map; (ii) the input to the SSM is mapped through an additional linear layer to recover the *values* in the linearized version of Attention; (iii) the SSM hidden state is expanded to accommodate for normalization terms, as per (10).

Finally, in Lst. 3 we report also our implementation of the Hedgehog projection operator, used within the HedgeMamba layer. Like in the original Hedgehog paper, the output of the feature map $\phi$ (6), is duplicated by collating its opposite. As a nonlinearity, we apply a softmax operation *along the embedding dimension*, instead of vanilla exponentiation as done in Zhang et al. (2024): this is to guarantee better numerical stability.

```python
# --- hedge-mamba mixer module --- #
class HedgeMambaMixer(nn.Module):
    def __init__(self, config):
        super().__init__()
        self.hidden_size_per_head = config.hidden_size // config.num_attention_heads

        # from mamba
        # state_size == hidden_size to mimic attention
        self.gate_proj = nn.Linear(config.hidden_size, config.hidden_size)
        self.conv1d = nn.Conv1d(config.hidden_size, hidden_size)
        self.x_proj = nn.Linear(
            config.hidden_size, config.time_step_rank + config.hidden_size * 2
        )
        A = nn.Parameter(self.init_A(config))
        self.dt_proj = nn.Linear(config.time_step_rank, self.hidden_size_per_head)
        self.out_proj = nn.Linear(config.hidden_size, config.hidden_size)

        # additional projections to replicate linear attention
        self.v_proj = nn.Linear(config.hidden_size, config.hidden_size)  # values
        self.hhog_q = HedgehogProjection(config, self.hidden_size_per_head)  # hedgehog
        self.hhog_k = HedgehogProjection(config, self.hidden_size_per_head)
        self.rotary_ndims = int(self.hidden_size_per_head * config.rotary_pct)
        self._init_rope()  # rope positional encoding

    def forward(self, hidden_states: torch.Tensor) -> torch.Tensor:
        gate = F.silu(self.gate_proj(hidden_states))
        hidden_states = self.conv1d(hidden_states)

        # linear proj to recover SSM parameters
        dt, B, C = torch.split(self.x_proj(hidden_states),
                               [time_rank, state_size, state_size],
                               dim=-1)

        # apply hedgehog feature map
        B = self.hhog_k(B.view(batch_size, seq_len, num_heads, hidden_size_per_head))
        C = self.hhog_q(C.view(batch_size, seq_len, num_heads, hidden_size_per_head))

        # rope positional encoding as in pythia
        C = self.rotary_emb(C, seq_len=C.shape[1])  # equivalent to Q from attention
        B = self.rotary_emb(B, seq_len=B.shape[1])  # equivalent to K from attention

        # value projection
        V = self.v_proj(hidden_states)

        # duplicate for score normalization as in attention
        A = torch.cat([self.A, self.A], dim=0)
        dt = torch.cat([dt, dt], dim=1)
        V = torch.cat([V, torch.ones_like(V)], dim=1)

        # leverage Mamba SSM mixer
        scan_outputs = selective_scan_fn(V, dt, A, B, C)

        # apply normalization
        scan_outputs = scan_outputs[:, : self.hidden_size_per_head, :] / \
                       scan_outputs[:, self.hidden_size_per_head :, :]

        # gate
        scan_outputs = scan_outputs * gate
        return self.out_proj(scan_outputs)
```

Listing 2: PyTorch-style pseudocode of our `HedgeMamba` sequence mixer, which equips the vanilla Mamba SSM mixer with the Hedgehog feature map (Zhang et al., 2024) for Attention linearization.

```
# --- hedgehog projection module --- #
class HedgehogProjection(nn.Module):
    def __init__(self, config, head_size, bias=True):
        super().__init__()
        self.config = config
        self.phi = nn.Linear(head_size, head_size, bias=bias)

    def forward(self, x: torch.Tensor) -> torch.Tensor:
        # x.shape: [B, S, H, D]
        #
        # B: batch size
        # H: number of heads
        # S: sequence length
        # D: per head embedding size
        x = self.phi(x)

        # negative mapping enabled as in hedgehog
        x = torch.cat([x, -x], dim=-1)  # [B, H, S, 2D]

        # NOTE: we use softmax as activation function here instead of
        # default exponential following hedgehog paper appendix to
        # avoid numerical overflows; softmax is applied on embedding
        # dimension here NOT sequence length as in standard softmax attention
        return x.softmax(dim=-1)
```

Listing 3: Implementation of the Hedgehog projection layer for Softmax Attention linearization (see Sec. 3.1 for details).

## D    USE OF LLMs FOR WRITING

We acknowledge the use of large language models (LLMs) to refine the writing and presentation of this paper. These tools were exclusively employed for grammatical correction, stylistic improvements, and overall polishing of the text. All original content, ideas, and research presented herein were conceived and developed solely by the authors.

