# OpenReview forum: "Attention to Mamba: A Recipe for Cross-Architecture Distillation"
_ICLR.cc/2026/Conference — Submitted to ICLR 2026_

### Official Review · Reviewer_jYaM · 2025-10-26

**Soundness:** 2
**Presentation:** 2
**Contribution:** 2
**Rating:** 4
**Confidence:** 4

**Summary:**

This paper aims to present a better strategy for distilling Transformers to Mamba. The major concept of this paper is introducing an intermediate step that transfers knowledge from softmax based Transformers to the linear attention variant, which is basically proposed by Hedgehog. Then, reusing the QKV weights to their corresponding CBX components in Mamba, and distilling it again, the authors show that it leads to better performance than previous works.

**Strengths:**

The paper is very straightforward -- discovering the importance of the intermediate Hedgehog transfer. As the idea is very simple, it's easy to understand, probably also for people who are not familiar with distillation or SSM literature.

**Weaknesses:**

As the idea is very simple, the novelty is limited. Most of the contents in the paper are about preliminary papers; for example, Section 1.1, Section 2 are only about previous works, and about 80% of Section 3 are introducing the ideas from Hedgehog or Mamba2.

**Questions:**

One critical ablation is missing: how important are "Parameter initialization" and "Attention scores normalization" in Section 3.2? Also, I am not convinced with the necessity of attention scores normalization. Mamba architecture does not have a separate normalization of such scores, but why did the authors choose to use the trick?

The paper only compares to Hedgehog. How is it compared to MOHAWK?

In Table 1, Hedgehog's PPL is 14.89. But in Table 3, the 100/0 row shows very high PPL. Where does this significant difference come from?

---

> ### Author Response · Authors · 2025-11-28
>
> >W1: As the idea is very simple, the novelty is limited. Most of the contents in the paper are about preliminary papers; for example, Section 1.1, Section 2 are only about previous works, and about 80% of Section 3 are introducing the ideas from Hedgehog or Mamba2.
>
> We agree with the reviewer that the main contribution in our paper does not lie in the proposed attention linearization approach, in that it directly expands on the Hedgehog work: as such, we believe that including a proper introduction to the ideas leveraged in our work is necessary to make the paper self-contained. Regardless, we do offer additional relevant and valuable contributions in our work, particularly:
>
> - We provide a principled way to use Hedgehog as a bridge to connect softmax attention with other sub-quadratic attention alternatives (with a focus on Mamba), to further boost its expressivity
> - We provide an extensive empirical analysis of the proposed distillation procedure, including ablations on model scaling, training token availability, and their distribution
>
> We believe these represent valuable contributions in their own right, especially given the breadth of the analyses involved.
>
> ---
>
> > Q1: One critical ablation is missing: how important are "Parameter initialization" and "Attention scores normalization" in Section 3.2? Also, I am not convinced with the necessity of attention scores normalization. Mamba architecture does not have a separate normalization of such scores, but why did the authors choose to use the trick?
>
> As the reviewer correctly points out, normalization is not necessary for the sake of stabilizing the architecture (indeed Mamba doesn’t use it); however, it’s useful for the sake of approximating the attention layer (which is the purpose of our work). Since stage 1 revolves around mimicking softmax attention, and softmax attention applies score normalization, we make sure to equip the attention linearization procedure with the capability of performing normalization: this is in line with the original Hedgehog paper (see Eq4 there), and in preliminary results it showed to consistently boost performance. For reference, without normalization we only achieve a perplexity of 55.68 at the end of stage 1, while including it we can reach ~26 (see also Fig3).
> Regarding the role of parameter initialization: not including this would correspond to performing naive distillation directly onto Mamba. Previous literature warns against this (see Wang et al, Bick et al, among our paper’s references), and our preliminary results in this sense were also disappointing, showing perplexities in the hundreds.
>
> ---
>
> > Q2: The paper only compares to Hedgehog. How is it compared to MOHAWK?
>
> Comparison with MOHAWK is challenging due to the different set-ups considered in the two papers: as a teacher model, we are considering a Pythia architecture (vs Phi) and as a training dataset we consider OpenWebText (vs C4). Adapting MOHAWK to the different setup (or vice-versa) would require a relevant effort in terms of computational resources. We believe that Hedgehog (which we compare against) represents the most relevant baseline, as our method directly builds upon it. Nonetheless, we point out that our method is supported by additional strengths over MOHAWK: particularly, its simplicity (by comparison, MOHAWK uses a three-staged approach where the relevant modules are progressively aligned), and the theoretical grounding on Mercer’s theorem.
>
> ---
>
> > Q3: In Table 1, Hedgehog's PPL is 14.89. But in Table 3, the 100/0 row shows very high PPL. Where does this significant difference come from?
>
> We appreciate the reviewer’s attention to detail. The numbers refer to different experiment set-ups: Tab1 reports the results from the original Hedgehog recipe (alignment + finetuning, with 50/50 training token split between the two), while Tab3 ablates over the distribution of training tokens between the two stages in our recipe. In particular, the 100/0 row corresponds to investing all the training tokens into the alignment stage: as such, it corresponds to vanilla Hedgehog (as there is no stage 2, we never include the Mamba components), without any finetuning. This explains the poor performance of this rather extreme setup.

---

### Official Review · Reviewer_g5YJ · 2025-10-28

**Soundness:** 3
**Presentation:** 4
**Contribution:** 2
**Rating:** 2
**Confidence:** 4

**Summary:**

The paper addresses the challenge of distilling a pretrained Transformer (Pythia) into Mamba to achieve faster inference and lower memory usage while retaining performance. Specifically, it proposes a two-stage distillation recipe: 1. Distilling the Transformer’s softmax Attention into a Linear Attention based on the Hedgehog approach. 2. Using this linearized Attention to initialize and fine-tune a Mamba-based architecture (HedgeMamba). The method is evaluated on the OpenWebText dataset, and the proposed HedgeMamba achieves superior performance compared to the simple direct distillation into Mamba and Hedgehog distillation into Linear Attention.

**Strengths:**

- The two-stage distillation process is novel and reasonable. It works much better than the simple direct distillation from Transformer to (Hedge)Mamba. Also, thanks to the superior expressiveness of Mamba over that of Linear Attention, the distilled HedgeMamba works better than Linear Attention models.
- The ablation studies effectively show the importance of the two-stage recipe.
- The paper is well written and provides enough details about their implementations, including the limitations.

**Weaknesses:**

#### Major Weaknesses
- The important information about the efficiency of HedgeMamba architecture is missing. Because the motivation of the distillation in this paper is to create an efficient model, the author should compare the inference speed or FLOPs. I doubt the efficiency of the HedgeMamba because the hidden stage dimension is huge (2048) compared to Mamba (16 to 64), in addition to the newly introduced layers.
- Why is the large SSM hidden state used? Although it is related to the previous weakness point, the inference speed should be much slower compared to the original Mamba (I guess 2~5 times slower) due to the large hidden state size. Did you use Mamba2 instead of Mamba? (Mamba2 should be relatively faster even if the hidden state size is large.)
- A gap exists between the motivation and the evaluation. Although the motivation is to borrow the strong performance of large Transformers, the experiments are conducted with the 1B size model. The Pythia-1B is inferior to Mamba-790M, and, to this end, HedgeMamba is inferior to Mamba-790M. In addition, the distillation cost (12 days with 8xA100 GPUs) seems not so small compared with the scratch training cost of Mamba.
- The lack of comparison against other methods proposing the Transformer to Mamba distillation [1, 2]. Although the authors mentioned that the experimental setups are different, comparisons can be possible by trying previous methods with this paper’s setup. Different architectures can be compared with accuracy-efficiency trade-off.

#### Minor Weakness
- Typo
    - L037) tokens representations -> token representations

[1] Wang, J., Paliotta, D., May, A., Rush, A., & Dao, T. (2024). The mamba in the llama: Distilling and accelerating hybrid models. Advances in Neural Information Processing Systems, 37, 62432-62457.
[2] Bick, Aviv, et al. "Transformers to ssms: Distilling quadratic knowledge to subquadratic models." Advances in Neural Information Processing Systems 37 (2024): 31788-31812.

**Questions:**

Please see major weakness. As to the third point, if it is difficult to evaluate with large models, it can be interesting if you can show some evaluations that the distillation improves the weak points of the vanilla Mamba described in some papers such as [3, 4]. In addition, if HedgeMamba is efficient, I want to raise the rating.

[3] Park, J., Park, J., Xiong, Z., Lee, N., Cho, J., Oymak, S., ... & Papailiopoulos, D. (2024). Can mamba learn how to learn? a comparative study on in-context learning tasks. arXiv preprint arXiv:2402.04248.
[4] You, W., Tang, Z., Li, J., Yao, L., & Zhang, M. (2024). Revealing and Mitigating the Local Pattern Shortcuts of Mamba. arXiv preprint arXiv:2410.15678.

---

> ### Author Response · Authors · 2025-11-28
>
> > W1: The important information about the efficiency of HedgeMamba architecture is missing. Because the motivation of the distillation in this paper is to create an efficient model, the author should compare the inference speed or FLOPs [...]
>
> The reviewer raises a fair point; however, the efficiency justification stems directly from the inner workings of the Transformer vs an RNN architecture. The FLOPs cost of performing autoregression in a Transformer model scales with sequence length, while for an RNN architecture it remains constant per new token generated. While the Hedgehog approximation we’re relying on does require a larger state dimension, operations across it are perfectly parallelizable. Unfortunately, for optimization purposes, Mamba’s `selective_scan` implementation caps the maximum hidden state dimension to 256 (see also issue #120 in Mamba’s repo). As a consequence, in our code we had to sequentialize its application across the hidden state dimension to be able to keep using Mamba’s fused kernel implementation (we point this out in footnote 2 in page 7). For this reason, timing figures are distorted and a direct comparison is unfaithful.
>
> ---
>
> > W2: Why is the large SSM hidden state used? [...]
>
> The large hidden state is necessary to match the Transformer’s dimensionality: notice we need to collect queries and keys from the Transformer, and further push them to a larger dimensionality to apply Hedgehog, before reducing them again. Vanilla Mamba, on the other hand, does not have such constraints, and can choose its dimensionality freely. Notice imposing this constraint is necessary for our purpose of minimally affecting the overall Transformer architecture: we aim to only replace the Attention mechanism, while leaving the MLPs untouched for the largest part.
>
> ---
>
> > W3: A gap exists between the motivation and the evaluation. [...]
>
> We refer to footnote 2 on page 7 regarding the total training times reported: we had to sequentialize some inherently parallel operations to be able to still use Mamba’s CUDA implementation of the fused `parallel_scan` kernel. This results in an inflated training time, which can be reduced by directly adapting Mamba’s CUDA implementation. A more faithful metric to reflect distillation cost is the number of training tokens used (10B vs 334B of the original model).
> Regarding the choice of models, we focused on Pythia as a candidate Transformer architecture in light of its transparency in making available information regarding its whole training procedure, which also explains its broad adoption in the open source and research community. We realize it’s not the most performant Transformer architecture available, but we believe it’s a reasonable reference point.
>
> ---
>
> > W4: The lack of comparison against other methods proposing the Transformer to Mamba distillation [...]
>
> While we agree with the reviewer that providing direct comparison with MOHAWK and LoLCATs baselines would help better positioning our paper, we highlight that this poses significant challenges, both in terms of the engineering effort required to setup a fair comparison, and in terms of the computational resources necessary for running exhaustive experiments. This is due to the stark differences in the setups (backbone models / datasets / goals) considered:
>
> - **LoLCATs** focuses on an instruction-tuned distillation process: it uses a masked QA loss over the Alpaca-cleaned dataset, to train the student on instruction/response pairs. Conversely, our method targets general-purpose pretraining: as such, it relies on text corpora such as OpenWebText. Any attempt to adapt these methodologies to different corpora would likely degrade performance, muddying the comparison.
>
> - Our purpose is more aligned with **MOHAWK**, but even in this case there exist relevant differences both in terms of backbone models (Pythia vs Phi) and of datasets (OpenWebText vs C4) considered. Crucially, in their repo the authors don’t make available their training pipeline, which forces us to conduct a full hyperparameter sweep to identify the optimal training configuration and ensure a fair comparison.
>
> For these reasons, we decided to focus mainly on Hedgehog as a baseline, given how our approach directly builds upon that work. In our opinion, this provides the most relevant comparison to gauge the improvements granted by our adaptations. While direct comparisons with other baselines proved challenging, we believe our work makes up for this by offering a method which is simpler and more theoretically grounded than the three-stage progressive alignment prescribed by MOHAWK. This simplicity, combined with the extensive ablations on model size, architectural components, and token budget allocations, provide a comprehensive insight into HedgeMamba’s robustness and effectiveness, and we believe constitutes a valuable contribution in its own right.

---

### Official Review · Reviewer_ET8B · 2025-10-30

**Soundness:** 3
**Presentation:** 3
**Contribution:** 2
**Rating:** 6
**Confidence:** 4

**Summary:**

The paper tries to fine-tune full attention into linear attention, first by using the Hedgehog distillation, and then fine-tune into a Mamba variant with parameters reused in the wake of state space duality. Empirical experiments and ablation studies verify the effectiveness and necessity of the two-stage recipe.

**Strengths:**

The idea of conversion to Mamba by reusing weights based on state-space duality is interesting.
Empirical results are good.

**Weaknesses:**

Stage 1 is identical to Hedgehog, which reduces the novelty of the approach.

**Questions:**

From my understanding, Stage 1 is identical to the Hedgehog work. Please clarify if there is any difference with their work. The identical part should be ideally presented in a more concise way, and details (e.g., the reference of kernelization/Mercer's theorem) should be put into the appendix, leaving more space for the novel part.

The difference between HedgeMamba and Mamba is important. It would be easier to follow if the author could prompt the reader earlier (e.g. in Fig 2) to check Fig 4 for it.

Have the authors considered the applicability to other linear attention approaches, e.g., DeltaNet?

Please also report the average accuracy in Table 1-3 for easier comparison.

The method becomes suspicious when the largest improvement comes from gated attention; Is the Mamba part really necessary? How about the case if you add gating only?

L69,350, ...: Should use \citep
L138,353,354, ...: Should use \citet
L249-253, ...: Should use "by \citet{...}" instead of "in \citet{...}"
Some numbers are out-of-margin in Table 1

---

> ### Author Response · Authors · 2025-11-28
>
> > Q1: From my understanding, Stage 1 is identical to the Hedgehog work. Please clarify if there is any difference with their work. The identical part should be ideally presented in a more concise way, and details (e.g., the reference of kernelization/Mercer's theorem) should be put into the appendix, leaving more space for the novel part.
>
> This is correct: stage 1 does follow the Hedgehog procedure. We included its description in the main text to make the paper self-contained, and accessible to readers who might be unfamiliar with the Hedgehog work. Notice Sec3.1 only requires half a page, so we believe it’s a worthy trade-off overall.
> However, we acknowledge this could be made more concise, and we’re happy to trim the discussion around Mercer’s theorem (L260-264), as per reviewer’s suggestion.
>
> ---
>
> > Q2: The difference between HedgeMamba and Mamba is important. It would be easier to follow if the author could prompt the reader earlier (e.g. in Fig 2) to check Fig 4 for it.
>
> We appreciate the reviewer’s suggestion, and agree that it will help improve clarity. To implement this, we will:
>
> - Include a reference in Fig2 caption, "See Fig4 for complete schematics of the architectures comparing HedgeMamba and vanilla Mamba."
>
> - Include a brief pointer in Sec3 when we first introduce HedgeMamba: "The key architectural differences between our HedgeMamba and vanilla Mamba are illustrated in detail in Fig4."
>
> ---
>
> > Q3: Have the authors considered the applicability to other linear attention approaches, e.g., DeltaNet?
>
> We thank the reviewer for raising an excellent point! Our focus on Mamba as target architecture was motivated by several factors, mainly:
>
> - Strong performance: Mamba arguably represents the state-of-the-art for sub-quadratic alternatives to attention
> - Theoretical connection: As discussed in Sec2.1 (and by Dao and Gu (2024)), Mamba has a direct mathematical correspondence with Linear Attention, allowing for a natural transition to stage 2 via the initialization strategy illustrated in Eq7.
>
> This being said, we believe our two-stage recipe could generalize to other linear attention approaches. For DeltaNet specifically, one would need to adapt the parameter mapping / initialization in stage 2 to account for its particular formulation of the recurrence relationship.
> We appreciate the reviewer’s suggestion, and will expand our Conclusions to include "While this work focuses on Mamba, we believe our two-stage approach could extend to other linear attention variants, such as DeltaNet, provided appropriate parameter mappings are established.", ensuring the reviewer gets acknowledged.
>
> ---
>
> > Q4: Please also report the average accuracy in Table 1-3 for easier comparison.
>
> We thank the reviewer for this practical suggestion that would improve the paper’s presentation. We will include an “Average” column in the tables, clarifying in the caption that some metrics are length-normalized accuracy while others are simply accuracy (as per Gu and Dao, 23).
>
> ---
>
> > Q5: The method becomes suspicious when the largest improvement comes from gated attention; Is the Mamba part really necessary? How about the case if you add gating only?
>
> We thank the reviewer for this thought-provoking question. Let us clarify that our goal is mainly to translate Attention into a Linear variant, and that we pick Mamba as the main representative for this class due to its demonstrated performance. Quantifying which specific component contributes most to Mamba’s expressivity is secondary to our goal. This being said, we understand the concern, but we believe that the collected results are actually in line with expectations, for two reasons:
>
> - The (+SSM) row in Tab2 refers to allowing fine-tuning of the SSM component of Mamba: these include its input-dependent parameters B, C, and the state matrix \Lambda. However, B and C already have a strong initialization coming from Stage 1’s Linearized Attention. The core additional flexibility allowed in this ablation would then mainly reduce to the simple learnable causal attention mask, \Lambda (see also L408)
>
> - Recent work [1, as well as the reported references in L415] highlights the role of the components outside the core SSM recurrence in boosting Mamba’s overall expressivity.
>
> Still, we appreciate the reviewer’s attention to detail, and we will further highlight their point in the main text.
>
> [1] Huang et al, Understanding input selectivity in Mamba, ICML 2025
>
> ---
>
> > Various typos
>
> We thank the reviewer for pointing out the typos. These have been corrected.

---

### Meta-Review · Area_Chair_tdoU · 2026-01-07

**Summary:**

The paper investigates the distillation of transformer models into SSMs. It proposes a two-stage method: first linearizing attention using the Hedgehog method, and then initializing a mamba variant using those weights followed by further distillation. While the reviewers find the approach straightforward and the empirical results on Pythia-1B respectable, the consensus leans towards rejection. The primary reasons are limited technical novelty, as the method heavily relies on existing techniques, and a significant concern regarding inference efficiency.

**Reviewer Concerns:**

Addressed part:

- Clarification of stage 1: The authors acknowledged that stage 1 is identical to Hedgehog but argued its inclusion is necessary for self-containment.

- Scaling and dataset choice: Authors justified using Pythia-1B and OpenWebText based on transparency and resource constraints.

- Ablation on normalization: The authors provided a clear explanation and preliminary numbers showing that attention score normalization is crucial for mimicking softmax behavior.

Outstanding part:

- Efficiency Paradox (Critical): Reviewer g5YJ raised a major concern about the hidden state dimension. In the rebuttal, the authors admitted that mamba’s optimized selective_scan kernel is capped at a dimension of 256. To support their 2048-dim state, they had to "sequentialize" the application, leading to distorted timing figures. This is a fundamental issue: if the distilled model cannot utilize fast SSM kernels, it fails the primary motivation of cross-architecture distillation.

- Lack of SOTA Comparisons: Reviewers jYaM and g5YJ noted the absence of comparisons with recent competitive baselines like MOHAWK or LoLCATs. The authors' rebuttal cited engineering difficulty and dataset differences, but empirical validation against the current state-of-the-art is expected.

- Incremental Novelty: Reviewers ET8B and jYaM both noted that 80% of the method is a combination of prior works (Hedgehog/Mamba2). The specific novel mapping/recipe was viewed as an incremental contribution.

**Reviewer Scores:**

Reviewer ET8B (Score: 6): Likely to decrease or maintain. While initially positive, the reviewer’s concern about whether the "mamba part is really necessary" vs. just adding Gated Attention was not fully dismissed by the rebuttal, and the efficiency issues raised by other reviewers would likely dampen their enthusiasm.

Reviewer g5YJ (Score: 2): Likely to stay at 2. The author's admission that their state dimension is too large for standard Mamba kernels confirms the reviewer’s suspicion that HedgeMamba is not actually efficient in practice.

Reviewer jYaM (Score: 4): Likely to stay at 4. The reviewer remained unconvinced by the novelty, and the rebuttal's explanation for the missing SOTA comparisons was a standard "effort-based" excuse that rarely elevates a score.

---

### Decision · Program_Chairs · 2026-01-26

Reject